# l-Alanine Exporter AlaE Functions as One of the d-Alanine Exporters in *Escherichia coli*

**DOI:** 10.3390/ijms241210242

**Published:** 2023-06-16

**Authors:** Satoshi Katsube, Keiichiro Sakai, Tasuke Ando, Ryuta Tobe, Hiroshi Yoneyama

**Affiliations:** 1Department of Cell Physiology and Molecular Biophysics, Center for Membrane Protein Research, School of Medicine, Texas Tech University Health Sciences Center, Lubbock, TX 79430, USA; 2Laboratory of Animal Microbiology, Department of Animal Science, Graduate School of Agricultural Science, Tohoku University, 468-1, Aramaki Aza Aoba, Aoba-ku, Sendai 980-0845, Japan; k-sakai@nibb.ac.jp (K.S.); tasuke.ando.d4@tohoku.ac.jp (T.A.); ryuta.tobe.c7@tohoku.ac.jp (R.T.); hiroshi.yoneyama.a4@tohoku.ac.jp (H.Y.)

**Keywords:** d-amino acids, d-alanine, exporter, AlaE

## Abstract

d-amino acids have recently been found to be present in the extracellular milieu at millimolar levels and are therefore assumed to play a physiological function. However, the pathway (or potential pathways) by which these d-amino acids are secreted remains unknown. Recently, *Escherichia coli* has been found to possess one or more energy-dependent d-alanine export systems. To gain insight into these systems, we developed a novel screening system in which cells expressing a putative d-alanine exporter could support the growth of d-alanine auxotrophs in the presence of l-alanyl–l-alanine. In the initial screening, five d-alanine exporter candidates, AlaE, YmcD, YciC, YraM, and YidH, were identified. Transport assays of radiolabeled d-alanine in cells expressing these candidates indicated that YciC and AlaE resulted in lower intracellular levels of d-alanine. Further detailed transport assays of AlaE in intact cells showed that it exports d-alanine in an expression-dependent manner. In addition, the growth constraints on cells in the presence of 90 mM d-alanine were mitigated by the overexpression of AlaE, implying that AlaE could export free d-alanine in addition to l-alanine under conditions in which intracellular d/l-alanine levels are raised. This study also shows, for the first time, that YciC could function as a d-alanine exporter in intact cells.

## 1. Introduction

Amino acids are structurally categorized into two enantiomers, L-amino acids and d-amino acids. Up to a half century ago, d-amino acids were thought to play a minor function in biological processes because d-amino acids were not present in living organisms, with the exception of bacteria and organisms, which use only L-amino acids as building blocks of ribosomally synthesized polypeptides [1]. With the advances of analytical technology, d-amino acids were proved to be present in extracellular environments [2] and a variety of organisms, including plants, mammals, and humans, and were found to play an important biological function in mammals, such as d-serine in the nervous system [3,4] and d-aspartic acid in the endocrine system [5].

The most well-known and comprehensively studied biomolecules containing d-amino acids in microorganisms are found in the cell walls, in which d-alanine (d-Ala) and d-glutamic acid (d-Glu) are the major components of peptidoglycan [6]. In addition, recent advances in analytical techniques have revealed that microorganisms appear to synthesize and utilize various other d-amino acids [7], and that millimolar levels of d-amino acids are released by diverse bacterial species in culture [8]. It has also been shown that certain d-amino acids are incorporated into cell walls and influence peptidoglycan strength toward osmotic shock [8] and that a mixture of several d-amino acids and the modulation of the c-di-GMP (bis-(3′-5′) cyclic dimeric guanosine monophosphate) signaling pathway prevent biofilm formation [9,10]. These observations suggested that d-amino acids exert a cell wall remodeling function in response to bacterial growth conditions [8,9] and function as signaling molecules in an ecosystem [11]. These interesting results raised the question of how bacteria secrete d-amino acids.

In terms of L-amino acids, the L-lysine exporter LysE in the soil bacterium *Corynebacterium glutamicum* was identified for the first time as a bacterial amino acid exporter. Since then, several L-amino acid exporters have been found in *Escherichia coli* and *C. glutamicum* [12,13,14,15,16,17,18]. However, no transporter that exports L-alanine has been found in bacteria until the identification of the L-Ala exporter AlaE [19]. In terms of d-amino acids, LysE was subsequently shown to export d-lysine, d-serine, d-arginine, and d-ornithine [20].

Previously, we discovered the L-alanine (L-Ala) exporter AlaE while studying the L-Ala metabolic pathway in *E. coli* [19]. We subsequently showed that (i) d-Ala partially inhibited the AlaE-mediated active transport of L-Ala in a dose-dependent manner through inverted membrane vesicle experiments [21] and that (ii) *E. coli* has one or more energy-dependent d-Ala export systems through intact cell experiments [22]. Based on these results, we hypothesized that AlaE could contribute to the d-Ala export activity of *E. coli* and as-yet-unknown transporter(s) could be involved in this activity. Therefore, in this study, we designed an experimental system to identify the transporters that allow *E. coli* to export d-Ala out of cells.

## 2. Results and Discussion

### 2.1. Rationale of the Experimental Design

We previously demonstrated that *E. coli* has one or more energy-dependent d-Ala export system(s) by employing the d-amino acid dehydrogenase (DadA)-deficient mutant strain JW1178 [22]. This mutant enabled the detection of the extracellular accumulation of d-Ala in the presence of L-alanyl–L-alanine (Ala-Ala) because JW1178 cannot catabolize d-Ala generated by alanine racemase by using Ala-Ala-derived L-Ala as a substrate, which leads to a relatively high level of intracellular alanine compared to that in the *dadA*-positive parent strain. We thus hypothesized that *dadA*-positive wild-type cells expressing a putative d-Ala exporter gene could also export d-Ala out of the cells in the presence of Ala-Ala. To test this, we selected 392 ASKA clones that harbored individual expression vectors of putative membrane proteins (Appendix A) based on the following criteria: (i) the presence of a signal sequence, (ii) the presence of membrane-spanning region(s), and (iii) the protein function has not been clarified previously.

### 2.2. Screening of Candidate Clones Exporting d-Ala by a Bioassay Approach

As a first step, we performed a bioassay in minimal agar medium [21] impregnated with the d-Ala auxotroph *E. coli* MB2795, which contains 6 mM Ala-Ala and 0.1 mM IPTG to induce putative membrane proteins harbored on each expression vector [23]. No growth halo of the indicator strain MB2795 around AG1, the parent strain of the ASKA clone, spotted on the surface of the screening medium was observed. In contrast, a growth halo of MB2795 was observed around many of the selected ASKA clones, a representative result is shown in Appendix A. Therefore, 65 clones were selected as candidates that may have the ability to export d-Ala (Table 1). However, it is interesting to note the poor growth of some ASKA clones under the conditions of this initial screening (Appendix A). We presumed that the reason for this poor growth may be the IPTG-induced strong expression of membrane proteins, which could negatively affect the growth of ASKA clones. Indeed, approximately half of the open reading frames cloned into the expression vector pCA24N showed a severe growth inhibitory effect upon induction by 1 mM IPTG [23]. In turn, this growth inhibition could cause cell lysis, leading to the release of d-Ala outside the cells. Consequently, the indicator strain MB2795 was able to grow. We next performed a second screening, in which the concentration of Ala-Ala was adjusted to 1 mM. Consequently, we selected 14 clones that supported the growth of MB2795 (Table 1).

We then performed a third screening, in which the concentrations of IPTG and Ala-Ala were set at 0.04 mM and 2 mM, respectively, given that physiologically relevant d-Ala exporters could export d-Ala, even at low intracellular concentrations. Accordingly, five genes (*yciC*, *ymcD*, *yidH*, *yraM* and *alaE*) were finally selected as candidate genes for encoding d-Ala exporters (Table 1).

### 2.3. Measurement of d-Ala Export Activity of Candidate Proteins for d-Ala Exporter

If the five d-Ala exporter candidate genes selected in the above experiments encode d-Ala exporters, the intracellular levels of d-Ala in the transformants expressing plasmid-encoded *alaE*, *yciC*, *yidH*, *YmcD* and *yraM* genes, host cells of which are JW2645, JW1267, JW3652, JW5133, and JW3116, respectively, are expected to be lower than that in the respective host strain (Table 2). The intracellular accumulation of [^3^H]d-Ala in the transformants JW3652/pCA24N-*yidH* and JW3116/pCA24N-*yraM* had almost the same intracellular d-Ala levels as that of the respective host cells, JW3652 and JW3116, respectively (Figure 1). In contrast, YciC- and YmcD-overexpressing cells showed a clear decrease in intracellular d-Ala levels as compared to those in the host cell (Figure 1). Notably, the intracellular levels of d-Ala obtained in the AlaE-overexpressing cells were found to be much lower those in the host cell. Furthermore, AlaE-mediated intracellular d-Ala levels were lower than those mediated by YciC and YmcD (Figure 1). In other words, the extent of d-Ala export observed in JW2645/pCA24N-*alaE* is larger than those in JW1247/pCA24N-*yciC* and JW5133/pCA24N-*ymcD*. It is worth noting that the intracellular levels of d-Ala observed in AlaE- and YciC-expressing cells were more statistically significant than those of respective host strains, JW2645 and JW1247, respectively. These results are in good agreement with our earlier study demonstrating that *E. coli* has d-Ala export system(s) [22]. However, the contribution of these candidate transporters to the d-Ala export activity is not clear under the conditions tested as the expression levels of individual proteins in each test strain are not known.

The analysis of the membrane topology of YciC and YmcD using SOSUI revealed that six and one transmembrane regions were predicted for YciC and YmcD, respectively. AlaE was previously found to have four transmembrane regions. Thus, it is suggested that YciC, in addition to AlaE, could be a novel d-Ala exporter.

### 2.4. Accumulation of d-Ala in Intact Cells Expressing Plasmid-Borne alaE Gene

Based on the previous results (Figure 1), AlaE appears to be one of the major d-Ala exporter(s) in *E. coli*. We thus determined the impact of expression levels of AlaE on the intracellular accumulation of [^3^H]d-Ala. The AlaE-deficient variant JW2645 harboring the empty vector pCA24N accumulated d-Ala in a time-dependent manner, reaching 11.93 nmol/mg dry weight at 5 min (Figure 2a). When JW2645 cells were transformed with pCA24N-*alaE*, from which *alaE* is induced by IPTG, they were incubated in the presence of 0.25 mM IPTG, causing the intracellular d-Ala level to be reduced to almost zero. Interestingly, JW2645/pCA24N-*alaE* grown in the absence of IPTG-accumulated d-Ala, reaching 8.57 nmol/mg dry weight at 5 min, which is approximately 30% less than that in JW2645/pCA24N, suggesting that a low level of *alaE* was expressed even without IPTG induction. When the *alaE* gene was induced in the presence of 0.005 mM IPTG, d-Ala accumulated at a level between that which was obtained in the absence and presence of IPTG (0.25 mM), reaching 3.57 nmol/mg dry weight at 5 min. These results indicate that AlaE can export d-Ala in an expression-dependent manner. We then analyzed the expression of AlaE using Western blotting. Although AlaE was not detected in the presence of 0.005 mM IPTG, a very faint signal was detected in the presence of 0.01 mM IPTG, and the expression became almost saturated with 0.1 mM IPTG (Figure 2b). These results clearly indicated that AlaE exports d-Ala in an expression-dependent manner.

### 2.5. alaE Overexpression Promotes d-Ala Export

To substantiate the d-Ala export activity of AlaE, we determined the extracellular d-Ala levels in JW1178, which lacks the *dadA* gene encoding d-amino acid dehydrogenase but harbors the *alaE*-expression vector pCN24N-*alaE* or pCA24N in the presence of 6 mM Ala-Ala. By employing this system, we could test whether AlaE actually exported d-Ala [22]. Control JW1178/pCA24N cells transported d-Ala out of the cells in a time-dependent manner, reaching an extracellular d-Ala concentration of approximately 0.6 mM after incubation for 2 h, as reported previously (Figure 3) [22]. We assume that this basal level of d-Ala export activity could be caused by chromosomally encoded AlaE and YciC, as described above. When *alaE* expression was induced by IPTG (0.1 mM), the extracellular concentration of d-Ala, as expected, increased along with the incubation time, and reached approximately 1.0 mM (Figure 3). These results clearly indicate that AlaE has sufficient activity to export d-Ala in addition to L-Ala.

### 2.6. Impact of alaE Overexpression on the Growth of dadA-Deficient E. coli Cells in the Presence of d-Ala

In our earlier study, we found that AlaE plays a pivotal role in avoiding the accumulation of a toxic level of L-Ala in intact cells and functions as a ‘safety valve’ in a peptide-rich environment, such as the animal intestine [26]. Therefore, we raised the question of whether the d-Ala export activity of AlaE may be physiologically relevant.

To address this possibility, we determined the impact of *alaE* overexpression on the growth of JW1178 lacking the d-Ala catabolizing enzyme DadA in the presence of d-Ala (Figure 4). When JW1178/pCA24N was grown in the presence of 90 mM d-Ala, its growth was inhibited compared with its parent strain BW25113, indicating that the high accumulation of intracellular d-Ala owing to the inability to catabolize d-Ala causes the growth inhibition of JW1178/pCA24N (Figure 4a). In contrast, the growth of JW1178/pCA24N-*alaE* was restored under the same conditions; the level of growth at 48 h was between that of BW25113 and JW1178/pCA24N (Figure 4a). The growth of all strains was similar in the absence of d-Ala (Figure 4b). The results can be interpreted to show that AlaE in conjunction with other d-Ala exporters, such as YciC identified in this study, could relieve stress caused by a high accumulation of intracellular d-Ala via their d-Ala export activity.

In addition to a diverse array of biological functions of d-amino acids in microbial ecosystems [11], various d-amino acids have been found to be present in the animal intestine [27,28]. Although these detected d-amino acids are derived from intestinal microbiota [27,28], the pathways through which free d-amino acids are secreted remain unclear. Recent studies have shown the importance of the free d-amino acids secreted from the intestinal microbiota as modulators of host–microbe interactions, which leads to the maintenance of healthy intestinal conditions via the animal’s immune system [28,29].

In conclusion, we demonstrated the experimental evidence that showed, for the first time, that AlaE can function as an d-Ala exporter in *E. coli*. In addition, hitherto uncharacterized membrane protein YciC could also export d-Ala. In our earlier study, we clearly demonstrated that *E. coli* has an energy-dependent d-Ala export pathway(s) [22]. Thus, further in-depth characterization of these d-Ala exporters from a viewpoint of their physiological activities provides a basis for understanding the as-yet-unknown questions: what is the physiological function of d-amino acid secretion in bacteria, and are d-amino acids secreted through their exporter(s) physiologically relevant for the host-microbe interaction?

## 3. Materials and Methods

### 3.1. Bacterial Strains and Plasmids

*E. coli* strains and plasmids used in this study are listed in Table 2. Cells were grown under aerobic conditions at 37 °C in L-broth containing 1% (*w*/*v*) tryptone, 0.5% (*w*/*v*) yeast extract, and 0.5% (*w*/*v*) NaCl (pH 7.2) or minimal medium containing 22 mM glucose, 7.5 mM (NH_4_)_2_SO_4_, 1.7 mM MgSO_4_, 7 mM K_2_SO_4_, 22 mM NaCl, and 100 mM sodium phosphate (pH 7.1) [30]. Kanamycin (KM) and chloramphenicol (CP) were added to the medium as needed.

### 3.2. Bioassay Screening for Identification of d-Ala Exporter Candidates

From the ASKA clone(-) library, which comprises 4122 genes with distinct expression vectors of *E. coli* genes [23], we selected 392 genes that were presumed to encode membrane proteins of unknown function using the database [31]. To evaluate the d-Ala export activity of the selected clones, bioassays were performed using *E. coli* MB2795, a d-Ala auxotroph, as the indicator strain. For this, 1 mL of MB2795 cells grown in LB-broth containing 50 µg/mL d-Ala at 37 °C overnight was collected by centrifugation (13,700× *g*, 5 min, 23 °C), washed twice with 1 mL of 0.85% NaCl, and resuspended in 1 mL of 0.85% NaCl. We then inoculated MB2795 cells (final cell density, ~10^7^ cells/mL) into prewarmed L-agar (1.5% *w*/*v*) containing 0.1 mM IPTG and 6 mM Ala-Ala, and solidified the medium. Next, 5 µL of the overnight-grown cells of each 392 ASKA clone was spotted onto the screening plate and incubated overnight at 37 °C. We selected candidate clones that showed a growth halo of MB2795 in the first screening. We repeated this protocol for the second and third screenings using adjusted concentrations of IPTG and Ala-Ala, of 0.1 mM IPTG/1 mM Ala-Ala and 0.04 mM IPTG/Ala-Ala 2 mM, respectively (Table 1).

### 3.3. d-Ala Accumulation in Intact Cells

The selected *E. coli* strain candidates, JW2645, JW1247, JW3652, JW5133, and JW3116, and their transformants harboring the corresponding expression plasmid, pCA24N-*alaE*, pCA24N-*yciC*, pCA24N-*yidH*, pCA24N-*ymcD*, and pCA24N-*yraM*, respectively, were grown in minimal medium containing 12.5 µg/mL each of chloramphenicol (CP) and kanamycin (KM) as needed overnight at 37 °C. Each strain was suspended in the same medium, and the cell density was adjusted to achieve an A_660_ of 0.15–0.2 and grown at 37 °C until A_660_ reached 0.6. Then, IPTG (final concentration, 0.01 mM) was added, and cells were incubated at 37 °C for 1 h. Next, cells were harvested by centrifugation (8700× *g*, 10 min, 4 °C), washed twice with 100 mM potassium phosphate (pH 7.5), and suspended in an assay buffer containing 100 mM potassium phosphate (pH 7.5) and 10 mM MgSO_4_. The cell density was adjusted to A_660_ of 1.25, and the cell solution was kept on ice until use. The reaction was initiated by adding 10 µL of the same buffer containing 100.2 µM [^3^H] d-Ala (specific activity, 10 mCi/mmol) to 40 µL of the cell suspension. After incubation at 23 °C for 1, 2, 3, 4, and 5 min, the reaction was terminated by diluting the cells with 20 volumes of quenching buffer containing 100 mM potassium phosphate (pH 5.5), 100 mM KCl, and 10 mM MgSO_4_ [32]. Subsequently, the mixture was quickly filtered through a glass fiber filter (GF75, Advantec, Tokyo, Japan) and washed twice with the same buffer. The membrane filters were placed in vials and immersed in 6 mL of the scintillant Filter Count (Perkin-Elmer, Norwalk, CT, USA). After 24 h, the radioactivity was counted using a liquid scintillation counter SLC-5001 (Hitach Aloka Medical, Tokyo, Japan).

### 3.4. Growth Measurements

*E. coli* JW1178 harboring either pCA24N or pCA24N-*alaE* and their parent strain BW25113 was grown in 4 mL of L-broth containing 12.5 µg/mL each of CP and KM as needed at 37 °C overnight. Cells were collected by centrifugation (13,700× *g*, 5 min, 4 °C), washed twice with ice-cold 0.85% NaCl, and resuspended in the same original volume of the saline. Cells (50 µL) were inoculated into 5 mL of fresh minimal medium containing 90 mM d-Ala, 0.01 mM IPTG, and appropriate antibiotics, and then incubated with shaking (120 rpm) at 37 °C. Cell growth was measured by monitoring the absorbance at 660 nm (A_660_).

### 3.5. Western Blotting Analysis

JW2645/pCA24N-*alaE* cells were grown in minimal medium containing 12.5 µg/mL each of CP and KM. Subsequently, the cells were suspended in the same medium, and the cell density was adjusted to A_660_ of 0.15–0.2. Then, the cells were grown at 37 °C until A_660_ reached 0.6. Subsequently, IPTG (final concentration, 0.005, 0.01, 0.1 and 0.25 mM) was added, and cells were incubated at 37 °C for 1 h further. Next, the cells were harvested, washed twice with 50 mM Tris-HCl (pH 8.0), and suspended in the same buffer containing 1 mM phenylmethanesulfonyl fluoride (Wako Pure Chemical Industries, Osaka, Japan). The cell suspension was disrupted by a sonic oscillation cycle, consisting of a 15-s exposure and 45-s intermittent cooling, in an ice-bath using a Bioruptor UCD-250 (Cosmo Bio Co., Tokyo, Japan) at the maximum output level until the cell suspension became translucent. The extracts were centrifuged at 15,000× *g* for 20 min at 4 °C to remove unbroken cells and debris, and the supernatant was centrifuged again at 100,000× *g* for 1 h at 4 °C. The pellet fractions were suspended in 50 mM Tris-HCl (pH 8.0) containing 1% sodium dodecyl sulfate, mixed with an equal volume of a sample buffer [33], and the membrane fraction containing 20 or 50 µg of protein was analyzed using SDS-PAGE. After blotting on a polyvinylidene difluoride membrane (Millipore, Bedford, MA, USA), staining was performed with a mouse monoclonal antibody raised against the hexahistidine tag (MBL, Nagoya, Japan) and alkaline phosphatase-conjugated secondary antibody [34].

### 3.6. d-Ala Export Assay

d-Ala export activity was determined as described previously [22] with some modification. Fully grown cells (400 µL) were inoculated into 20 mL of fresh minimal medium containing 12.5 µg/mL KM and 12.5 µg/mL CP, and grown to the mid-log phase (A_660_ = 0.6). Cells were collected, washed twice with ice-cold minimal medium, suspended in the same medium, and adjusted to an A_660_ value of 3.0. The reaction was started by the addition of 6 mM Ala-Ala and 0.1 mM IPTG, and subsequent incubation at 37 °C. Cells (200 µL) were then collected by centrifugation (13,700× *g*, 5 min, 23 °C) after 10, 30, 60, and 120 min incubation. d-Ala concentrations in the supernatant samples were determined using high-performance liquid chromatography (HPLC) as described below.

### 3.7. Determination of an Extracellular d-Ala by HPLC

The concentration of d-Ala in the supernatant was determined as described previously [22]. Briefly, the supernatant was mixed with an equal volume of chloroform by vigorous shaking for 10 min and then centrifuged (12,000× *g*, 5 min, 23 °C). The resulting aqueous phase was mixed with ethanol (aqueous phase:ethanol = 4:1) and centrifuged as described above. Subsequently, 150 µL of the supernatant was mixed with 60 µL of 1 M NaHCO_3_ and 210 µL of 0.25% Nα-(5-fluoro-2,4-dinitrophenyl)-L-alaninamide (Tokyo Chemical Industry Co., Tokyo, Japan) followed by shaking in the dark at 37 °C for 90 min to derivatize D/L-Ala. After the neutralization of the solution with 60 µL of 1 M HCl, d-Ala was analyzed using HPLC using a column COSMOSIL/5C18-AR-II (Waters, Nacalai Tesque, Kyoto, Japan) to determine the absorption at 340 nm.

### 3.8. Determination of Protein Concentration

The protein concentration was determined using the method of Lowry et al. [35].

### 3.9. Statistical Analysis

The experiments were repeated three times as presented in each data. The data in this study are expressed as mean ± standard deviation. Statistical significance was calculated by two-tailed Student’s *t* test. Values of *p* < 0.05 were accepted as statistically significant in any analysis.

## Figures and Tables

**Figure 1 ijms-24-10242-f001:**
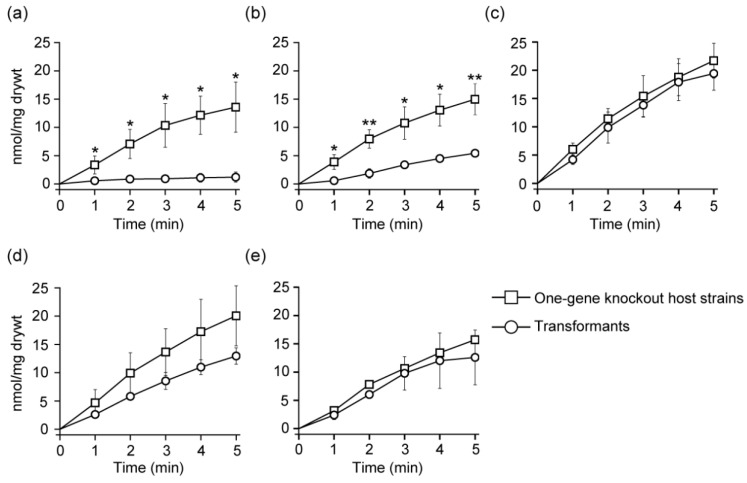
Accumulation of [^3^H]d-Ala in intact cells. Cells were grown in minimal medium and suspended in the same medium. The transport assay was initiated by the addition of [^3^H]d-Ala to a final concentration of 20 µM followed by incubation at 37 °C. An aliquot of the cell suspension was filtered through a membrane filter with a pore size of 0.22 µm and then washed twice with 3 mL of pre-warmed minimal medium. Values are the mean and standard deviation of results from three independent experiments. Symbols: □, one-gene knockout host strain (**a**) JW2645, (**b**) JW1247, (**c**) JW3652, (**d**) JW5133, and (**e**) JW3116; ○, transformants harboring respective expression plasmid (**a**) JW2645/pCA24N-*alaE*, (**b**) JW1247/pCA24N-*yciC*, (**c**) JW3652/pCA24N-*yidH*, (**d**) JW5133/pCA24N-*ymcD,* and (**e**) JW3116/pCA24N-*yraM*. * *p* < 0.05 and ** *p* < 0.005 by two-tailed Student’s *t*-test between the transformant and its host at each time point.

**Figure 2 ijms-24-10242-f002:**
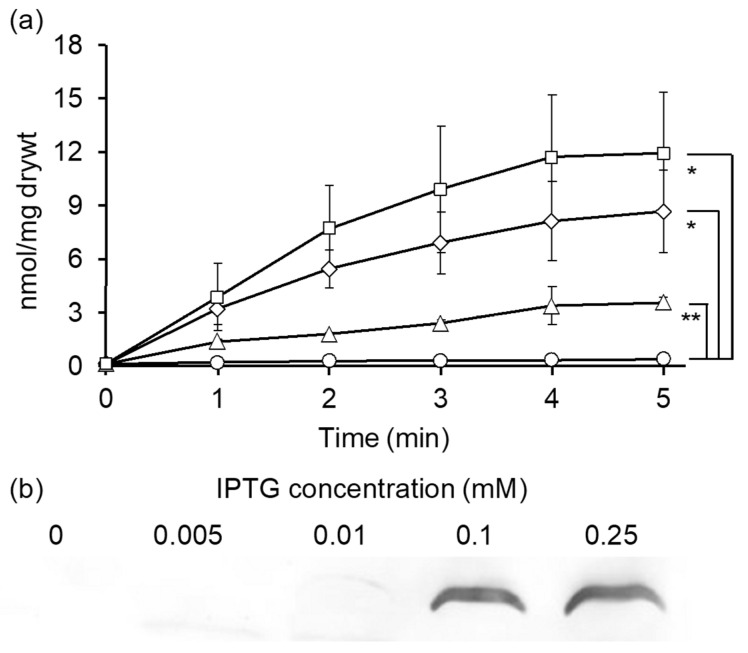
Accumulation of d-Ala in intact cells. (**a**) Cells of JW2645 strain with pCA24N or pCA24N-*alaE* were incubated with shaking at 37 °C in minimal medium and suspended in the same medium. The d-Ala accumulation assay was initiated by the addition of 20.04 µM [^3^H] d-Ala (specific activity) at a final concentration to the cells. The reaction was terminated by filtering through a membrane filter with a pore size of 0.22 µm, followed by two washes with 3 mL of quenching buffer. Values are the mean and standard deviation of results from three independent experiments. The following symbols were used: squares (JW2645/pCA24N), diamonds (IPTG 0 mM), triangle (IPTG 0.005 mM), and circles, JW2645/pCA24N-*alaE* (IPTG 0.25 mM). * *p* < 0.05 and ** *p* < 0.005 by two-tailed Student’s *t*-test at 5 min. JW2645/pCA24N-*alaE* (IPTG 0.25 mM) versus JW2645/pCA24N, *p* = 0.026; JW2645/pCA24N-*alaE* (IPTG 0.25 mM) versus JW2645/pCA24N-*alaE* (IPTG 0 mM), *p* = 0.025; JW2645/pCA24N-*alaE* (IPTG 0.25 mM) versus JW2645/pCA24N-*alaE* (IPTG 0.005 mM), *p* = 0.0041. (**b**) Expression of AlaE under various IPTG concentrations. Cells of strain JW2645/pCA24N-*alaE* were grown to a log phase. Then, an appropriate amount of IPTG was added to the medium and incubated for 1 h. The fraction containing membrane proteins was recovered, and the protein (0–0.01 mM IPTG was 50 µg, and 0.1–0.25 mM was 20 µg) was analyzed by SDS-PAGE and Western blotting.

**Figure 3 ijms-24-10242-f003:**
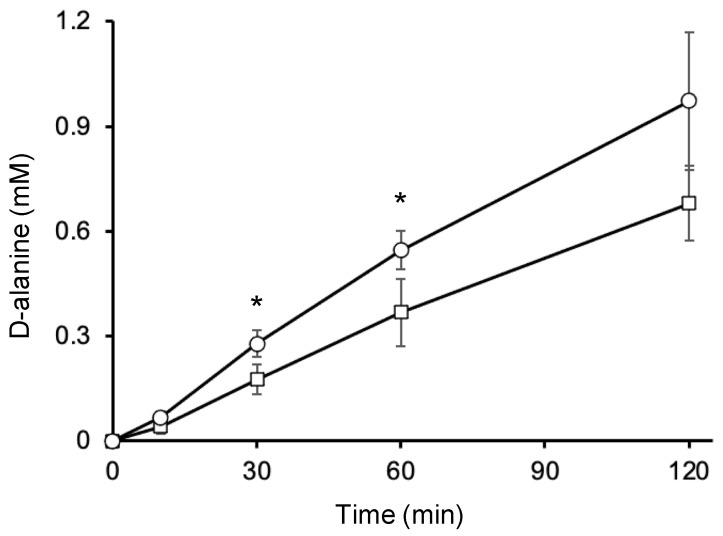
Secretion of d-Ala from *dadA*-disrupted *E. coli* cells. Cells of strain JW1178 with pCA24N (squares) or pCA24N-*alaE* (circles) were incubated in minimal medium in the presence of 6 mM Ala-Ala and 0.1 mM IPTG as needed, and the extracellular fractions were collected at 10, 30, 60 and 120 min. The d-Ala concentration in the supernatant was determined using HPLC. Values are the mean and standard deviation of results from three independent experiments. * *p* < 0.05 by two-tailed Student’s *t*-test. *p* values at 30 min and 60 min are 0.039 and 0.048, respectively.

**Figure 4 ijms-24-10242-f004:**
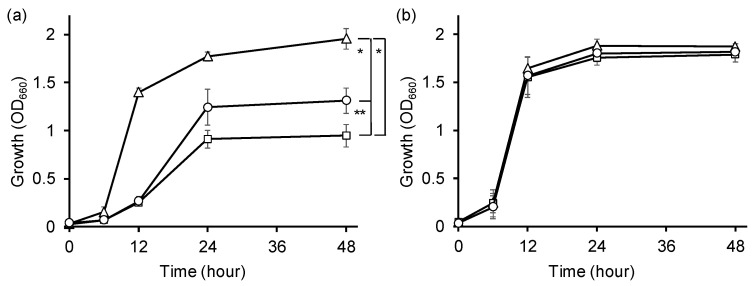
Growth (OD_660_) of *E. coli* strains in minimal medium in the absence of d-Ala or in the presence of d-Ala. Cells of strain JW1178 with pCA24N (squares) or pCA24N-*alaE* (circles) and their parent strain BW25513 (triangles) were grown in minimal medium in the presence of 90 mM of d-Ala (**a**) or in the absence of d-Ala (**b**) at 37 °C with shaking. Values presented are the mean and standard deviation of results from three independent experiments. * *p* < 0.05 and ** *p* < 0.005 by two-tailed Student’s *t*-test at 48 h. JW1178/pCA24N versus JW1178/pCA24N-*alaE, p* = 0.0026; JW1178/pCA24N versus BW25113, *p* = 0.016; JW1178/pCA24N-*alaE* versus BW25113, *p* = 0.042.

**Table 1 ijms-24-10242-t001:** Candidate ASKA clones selected by bioassay screening.

Screening	Conditions		Genes Selected
	Ala-Ala (mM)	IPTG (mM)	
First	6	0.1	*ybiP*, *ymcE*, *ycjF*, *ydeA*, *ydeS*, *yebE*, *ydhU*, *ymfR*, *yeaQ*, *yhjX*, *ynfA*, *ybbV*, *yfdH*, *yfdG*, *yfbW*, *yfbV*, *ygaM*, *ygdD*, *ygfX*, *yqjE*, *yraM, yhiP*, *yrbK*, *yidH*, *yifL*, *yibN*, *yjfL*, *yijD*, *yjeT*, *yohO*, *ymcD*, *yjbO*, *yqjF*, *yiaB*, *ykgB*, *yihF*, *ybjT*, *yjdB*, *yjeM*, *yohJ*, *ybdJ*, *ybhQ*, *ybhL*, *ybjO*, *ykgH*, *ybjM*, *ydgG*, *ycfZ*, *yciC, yebZ*, *yggT*, *yqiJ*, *yfeZ*, *yicL*, *yidG*, *yeiS*, *ylaC*, *yigG*, *ycdZ*, *ymfA*, *yfbJ*, *yohC*, *ynjI*, *yfdI*, *alaE*
Second	1	0.1	*yeaQ*, *yfbV*, *yqjE*, *yraM*, *yidH*, *yifL*, *yjeT*, *ymcD*, *yjbO*, *yihF*, *ybdJ*, *ykgH*, *yciC*, *alaE*
Third	2	0.04	*yciC*, *ymcD*, *yidH*, *yraM*, *alaE*

**Table 2 ijms-24-10242-t002:** Bacterial strains and plasmids used.

Strains and Plasmids	Characteristics	Reference
Strains		
*E. coli* BW25113	*rrn*B3, ∆*lacZ4787*, *hsdR514*, ∆(*araBAD*)567, ∆(*rhaBAD*)56	[24]
*E. coli* JW1178	∆*dadA::KM^r^* derived from BW25113	[24]
*E. coli* JW2645	∆*alaE::KM^r^* derived from BW25113	[24]
*E. coli* JW1247	∆*yciC::KM^r^* derived from BW25113	[24]
*E. coli* JW3652	∆*yidH::KM^r^* derived from BW25113	[24]
*E. coli* JW5133	∆*ymcD::KM^r^* derived from BW25113	[24]
*E. coli* JW3116	∆*yraM::KM^r^* derived from BW25113	[24]
*E. coli* AG1	F^-^ *endAl hsdRJ7* [r_k_^−^ m_k_^+^] *supE44 thi*-*J recAl gyrA96 relAl* X	
*E. coli* MB2795	d-Ala auxotroph (∆*alr*, ∆*dadX*) derived from MG1655	[25]
Plasmids		
pCA24N	CP^r^, *lacI*^q^	[23]
pCA24N-*alaE*	pCA24N harboring the *alaE* gene	[23]
pCA24N-*yciC*	pCA24N harboring the *yciC* gene	[23]
pCA24N-*yidH*	pCA24N harboring the *yidH* gene	[23]
pCA24N-*ymcD*	pCA24N harboring the *yraM* gene	[23]
pCA24N-*yraM*	pCA24N harboring the *ymcD* gene	[23]

## Data Availability

Raw data were generated at Tohoku university. Data supporting the findings of this study are available from the corresponding author S.K. on request.

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
