# Peer review of "l-Alanine Exporter AlaE Functions as One of the d-Alanine Exporters in Escherichia coli"

_ijms, 2023, doi:10.3390/ijms241210242_

Round 1
Reviewer 1 Report (Previous Reviewer 1)
The revised version of this manuscript addressed all comments adequately.
Reviewer 2 Report (Previous Reviewer 2)
I have no further comments.
Reviewer 3 Report (Previous Reviewer 3)
I have reviewed the revised manuscript and note that this has been substantially improved and all matters raised in the initial review have been addressed.
I recommend that the current version is reviewed by the authors and/or editorial staff to see if any typographical errors have been introduced in rewriting some of the sections - it was difficult to determine this in the marked-up version.
The English expression is clear and not confusing, although some phrasing could be improved - this is not a major issue.
This manuscript is a resubmission of an earlier submission. The following is a list of the peer review reports and author responses from that submission.
Round 1
Reviewer 1 Report
Katsube et al present a concise report documenting the identification of two E. coli genes that function as D-alanine exporters. Detailed work was done on the AlaE gene while the second gene, YciC, was only briefly examined. One would hope that the same group will follow this up in a future study. This manuscript is well written and all experiments are will described. I have only one small comment.
It might be helpful for readers to indicate the number of replicate experiments in the legend for Fig. 1. It appears that there are error bars for some of the overexpression experiments but it is not completely clear from the graphics that there were replicate experiments done because many points don’t show error bars. This was done, for example, in Fig. 3 (“Values are the mean and standard deviation of results from three 171 independent experiments.”)
Minor detail (abstract):
There is a difference in meaning between:
This study also shows that, for the first time, YciC could function as a D-alanine exporter…
and
This study also shows, for the first time, that YciC could function as a D-alanine exporter…
The authors likely meant the second version. The first version is unclear and suggest that YciC only functions as a D-alanine transporter for the first time and then stops.
Reviewer 2 Report
This study investigated the function of L-alanine exporter AlaE and reported that it might be one of the D-alanine exporters in Escherichia coli. Gnnerally, the manuscript is well designed and suitable for publication in the current journal. However, I recommend some further improvements before publication.
Introduction
1. add more recent references for "prevent biofilm formation", such as Liu et al., 2022, DOI: 10.1016/j.biotechadv.2022.107915.
2. Add one paragraph to discuss the chemical biology of D-amino acids as a molecular player before you start your topic.
3. Are there any reports on the function of L-alanine exporter AlaE? If so, discuss them ahead of the last paragraph.
Results and Discussion
1. The overall section is written like Results description without any discussion, especially for Section 2.1-2.5. The authors are strongly suggested strengthenning the dicussion based on the current reports.
2. Some statistical significant differences should be presented in the results.
3. In figure 3, why did you only detect the extracellular D-Ala levels in JW1178 for 120 min? How about 150 min or longer?
4. Provide a Short conclusion section at the end of R and D section.
5. L89-91, where are the five genes (yciC, ymcD, yidH, yraM, and alaE) from? Are there any reports on their functions? If so, please discuss them one by one. For example, the study by Bai et al., 2021, DOI: 10.1016/j.ijfoodmicro.2021.109189 has reported such details.
M & M
1. it is not applicable to write your methods by only citing others' methodology. The authors should carefully introduce how you carried out your experiment using the methodology. Please list the specific steps.
Reviewer 3 Report
In 2005, an expression library for the genome of E. coli was developed and this provides a resource for determining the function and biological role of yet unannotated genes. The authors call on this resource to address the question of how E. coli exports D-alanine, which plays a role in peptidoglycan synthesis/cross linking and possibly in cross-talk in the microbiome. Although D-Ala transporters in Gram positive systems (noted for C. glutamicum in the introduction) are known, the nature of the transports involved in E. coli for excretion of D-Ala have not been characterised, although prior publications from this group (2016, ref. 10) implicated the AlaE gene (which transports L-Ala) and potentially others. The current submission represents a substantive piece of research which identified candidate surface proteins in the genome library of E. coli that supported growth of a D-Ala indicator strain auxotrophic for this amino acid. The outcome was demonstrating that AlaE and a second protein, YciC, contribute to D-Ala transport, a useful novel finding.
The data in the manuscript is clearly presented but there are several areas where the phrasing can be improved, to eliminate confusion wrought through using abbreviated (short-cut) terminology. Many of these issues are marked on the annotated pdf file to show where and why some confusion arises. In brief:
1. Table 1 provides a list of strains and plasmids used in this study but this is not referred to in the text, so understanding which strains were hosting vectors at each experimental point is simply not clear. This is an inherent problem with the Materials/Methods coming after the Results/Discussion sections.
2. It is important that the figure legends refer to the methods sections which describe the full detail of the protocols used. Short-cut phrasing (e.g. ‘overexpressed’ and ‘deficient’, see Figs.) cannot be interpreted easily – what strain/plasmid combinations are used?
3. Some of the diagrams have very large error bars which overlap between tests and controls. Biological and/or technical replicates need to be clearly spelt out, as the former would lead to larger error than the latter.
4. The format of the Reference list needs to be checked for journal style – which would be addressed in final editing/review by the authors.
Overall:
This is a very useful, substantive piece of research which only lacks in the clarity of specifying which strains were used for each of the experiments described – there may be an error in describing the host/plasmid combination in the methods section and this requires review. The annotated pdf raises issues of potential confusion at several points.
